# "You have to stay in your house. . .because trouble can come": The impact of education, policy, and COVID-19 on menstruation experiences in Florence, Italy

Meagan R. DeMark[1], Grace A. Khamis [2], Megan S. Rach[3], Jaslin A. Martinez[4], Andrea L. DeMaria[4,5]*

1 Department of Speech, Language, and Hearing Sciences, College of Health and Human Sciences, Purdue University, West Lafayette, Indiana, United States of America, 2 Department of Biomedical Health Sciences, College of Health and Human Science, Purdue University, West Lafayette, Indiana, United States of America, 3 School of Nursing, College of Health and Human Science, Purdue University, West Lafayette, Indiana, United States of America, 4 Department of Public Health, College of Health and Human Sciences, Purdue University, West Lafayette, Indiana, United States of America, 5 March of Dimes, Arlington, Virginia, United States of America

* ademaria@purdue.edu

**Data Availability Statement:** Data relevant to this study are available from the Purdue University Research Repository at DOI:10.4231/81KK-3E74.

## Abstract

Starting from menarche to menopause, menstruators have an overall negative view of menstruation, and there is a prevalent need for community awareness to increase regarding this topic. Menstruators in Italy and Europe arrive at menarche earlier than they have in previous decades, allowing less time for them to receive preparatory knowledge. Some European countries have started employing menstrual policies, yet current menstruation education minimally supports menstruators in terms of recognizing irregular symptoms and learning how to manage them. Additionally, the COVID-19 pandemic caused menstrual lifestyle patterns and experiences to be altered. The purpose of this study is to gain an understanding of menstruation-related education and policies accessible to menstruators and the impact COVID-19 had on menstruators. Researchers conducted 28 in-depth interviews in English with menstruators aged 18+ years who lived in or around Florence, Italy. All interviews were transcribed verbatim. Researchers used thematic analysis during coding to explore and understand participants' menstruation education at menarche, menstrual leave policy opinions, and how COVID-19 impacted their menstruation experiences and the availability of menstrual resources. Participants discussed varying timelines regarding when they were taught about menstruation. Their education sources varied between family, peers, personal experiences, and school. Most participants expressed enthusiastic feelings toward menstrual leave policies. Many participants reported having faced challenges accessing menstrual products during COVID-19. Most participants expressed having positive impacts, while few expressed negative impacts during COVID-19. Results found most menstruators learned about menstruation through their family before menarche but did not consider themselves prepared for their first cycle. The COVID-19 pandemic influenced many menstruators to adopt some lifestyle habits, shedding light on the varying opinions on the need for a

**Funding:** This research was partially funded by the Purdue University Office of Programs for Study Abroad, International Programs (Study Abroad and International Learning Grant and Intercultural Pedagogy Grant), received by AL DeMaria. No additional external funding was received for this study.

**Competing interests:** The authors have declared that no competing interests exist.

menstrual leave policy. Revamped policies related to timing, education type, and personal leave should be considered.

## Introduction

In this paper, 'menstruators' refers to individuals who experience menstruation. Conversely, the term 'non-menstruators' refers to individuals who do not experience menstruation. These term aim to include diverse gender identities by acknowledging that not all who menstruate identify as female, recognizing the importance of language in affirming the experiences of all individuals within our research scope.

Menstruation, commonly called a 'period,' is the shedding of the uterine lining. Menarche, defined as the first instance of menstruation, differs for each individual [1]. On average, menarche occurs between the ages of 8 and 16, and menstruation continues until menopause, the last occurrence of menstrual bleeding [2], which occurs at the average age of 51.4 years old [3]. Menstruation is typically inconsistent and does not occur in a set timeframe directly after menarche; after a few years, the body will develop a regular menstrual cycle lasting 21 to 45 days [4]. Once regulated, the average bleeding time during a menstruation cycle is 4 to 6 days [5]. Many perceive menstruation and menopause negatively due to symptoms such as back pain, cramping and aching, bloating, hot flashes, and fatigue [6, 7]. Additionally, menstruators have negative views of menopause due to a poor relationship with the ideals of aging [8] and the end of their reproductive era [3]. This negative stigma is decreased through public health education [7], yet research suggests about 25% of menstruators do not have adequate supplies or information to protect and manage their experience [9].

Menstruation remains a taboo subject in Europe [10]. When menstruation is talked about, it is typically done so in private, preventing those affected from receiving the treatment or support they require [10]. European menstruators commonly report having heavy menstrual bleeding and either did not contact their physician or did not receive treatment for related symptoms [11]. Italian menstruators start their cycle earlier than in previous decades due to higher socioeconomic status and food availability, allowing them to reach the necessary body mass index to induce menarche [5]. With menarche starting at a younger age, there is a lack of education informing menstruators about the expectations surrounding menstruation and overall health [12]. Less than half of Italian menstruators prefer having a monthly cycle, with most preferring a frequency of three months or never [12].

Workplace policies exist to support menstruators experiencing adverse symptoms. Related symptoms such as cramps, nausea, and fatigue can be caused by menstrual illnesses such as endometriosis, which can affect one's ability to work and prospective reproductive health [6]. Under potential menstrual leave policies, menstruators experiencing negative symptoms may be granted excused absences from work [6]. The implementation of occupational menstruation policies varies across the world. In Italy, a menstrual leave policy has been discussed but has not currently been passed.

Since 2019, the Coronavirus (COVID-19), a relationship between high stress and anxiety with menstrual cycle irregularity [13] has emerged. In addition to stress caused by COVID-19, the COVID-19 vaccine was also found to affect a majority of Italian menstruators' length of cycle as well as the quantity of menstrual bleeding [14]. Furthermore, healthcare adjustments during the pandemic made it more difficult for individuals to access healthcare [15]. Primarily, COVID-19 has shown how services and healthcare resources were allocated to the more dire cases, creating challenges for those seeking non-emergent services, including those related to menstruation [16].

## Study purpose

Challenges to menstruation management increase individuals' vulnerability to disease and overall health and well-being [17]. Little research has explored menstruation experiences among menstruators living within or near Florence, Italy, especially through qualitative approaches. Understanding menstruation experiences before and during the COVID-19 pandemic is imperative for identifying menstrual changes presented due to this significant event. The purpose of this paper is to understand menstruation experiences among menstruators living within or near Florence, with a specific focus on participants' menstruation awareness at menarche, thoughts on menstrual leave policies, and the impact COVID-19 had on menstrual experiences and access to resources.

## Methods

As part of a broader research project, menarche through menopause in Florence, Italy, this study focuses on education at menarche, menstrual leave policy, and the effects of COVID-19 on menstrual experiences. Undergraduate researchers recruited 28 eligible participants and conducted interviews from May 16, 2022 to June 17, 2022. The parameters set for participation included being 18 years or older, speaking proficient conversational English, having experienced menstruation, and residing near Florence, Italy. Participants who did not meet the criteria were excluded from the study. Qualitative data were gathered to highlight and assess an individual's preliminary attitudes and experiences toward menstruation. This study was approved by the Purdue Institutional Review Board (IRB No: 2022–318) with a letter of support from Florence University of the Arts. The research conformed to all ethical principles for medical research on human subjects, per the Declaration of Helsinki.

Participants were mainly sought out through direct, in-person recruitment in the English language. Recruitment occurred at stores, cafes, public areas, and educational buildings within the city limits of Florence, Italy. Snowball sampling was also used, in which past participants were asked to recommend other eligible individuals for the study [18]. The base sample of interviewees used to help connect researchers with others interested in participating consisted of roughly one-third of the total participants. These initial contacts were menstruators whom researchers interacted with in the previously listed public areas, who helped recruit approximately another one-third of the participants. The researchers' contact information in the study was given primarily through the initial contact with the potential participants. Further communication used WhatsApp, text messaging, or email. To maintain confidentiality, an identification number was assigned to each participant. All information regarding the privacy and contact information of the participant was not correlated with the identification number. All participants signed an informed consent form before engaging in the audio-recorded interview. In addition to the consent form being signed before data collection, the first question asked during the audio recording was whether the interviewer had the participant's consent to record their conversation. Interviews lasted, on average, 48 minutes and 9 seconds in a meeting place agreed upon between both the researcher and participant. Using Otter.ai Pro, a paid subscription application for audio recording and transcribing, each interview was securely recorded with the addition of a protected server. Every participant who engaged in the study was offered 20 euros as compensation, and most accepted. Data collection continued until all aspects of the inquiry were satisfied, ensuring all necessary data was collected and fully developed.

Interviews were conducted using a semi-structured protocol allowing the researcher and participant the flexibility to adjust or include additional questions or concerns regarding their responses. This protocol allowed participants to further develop their opinions and

**Table 1. Interview topic questions.**

| Topic | Questions |
|---|---|
| Education | How do you feel menstruation education has changed over time in Italy?<br>How do menstruation experiences and attitudes differ between generations? |
| COVID-19 | How has the ongoing COVID-19 pandemic impacted your menstruation experiences?<br>If you received the COVID-19 vaccine, did you notice any differences in your period? |
| Policy | What types of policies have you heard about that support menstruators?<br>How has your menstrual cycle affected your work?<br>How would a policy allowing so many missed hours of work for individuals with menstrual cycles be helpful?<br>Are there places in your community that offer free pads, tampons, or other menstruation-related products?<br>How are menstrual products disposed of? |

perspectives about the topics of this research. The interview began with general questions regarding the participant's life and general health, which helped to build rapport and establish a comfortable environment [18]. The participants discussed their experiences and perspectives on the topic of menstrual education at the time of menarche, menstrual leave policies, and how COVID-19 impacted menstrual experiences. Interviewers wanted to better understand menstruators' knowledge of menstrual policies, menstrual education, familiarity with supplies, and differences in cycles during COVID-19 and after receiving the vaccine. These probing questions can be seen in Table 1.

## Research team

Twenty-eight undergraduate students from the United States collected and transcribed data during a study abroad research-based program. All students implemented graduate-level research methodologies through being engrossed in Florentine culture. More specifically, the comprehensive graduate-level training encompassed rigorous instruction in qualitative research methodologies, interview techniques, and subject-specific knowledge pertinent to the study's focus areas. This training equipped student-researchers with the necessary expertise to navigate the complexities of the interview process effectively and ensure the collection of high-quality data. Data reliability was monitored by the Principal Investigator (PI), the last author, for procedures and outcomes. Themes were identified through code manuals and mind mapping. Discussion of coding and themes occurred frequently throughout the research and inconsistencies were adjusted through unanimity.

## Data analysis

Data analysis for this paper was completed by four undergraduate researchers, who were supervised by the PI and have an extensive research background, specifically in women's reproductive and sexual health. An immersive, full content review was first completed to ensure familiarity with all data, noting immediate patterns and ideas for potential codes and themes [19, 20]. Following this, thematic analysis was employed while creating the codebook to foster a substantial representation of the data. Codes were first collected by using a deductive process and added to a preliminary codebook. The inductive aspect permitted the ability to alter codes to identify important themes. Utilizing HyperRESEARCH 4.5.1, the authors were able to conclude the coding procedure. Multiple coding rounds were completed to ensure accuracy. The creation of the codebook and the coding were completed in May-June 2022.

When the coding process was finalized, data were gathered into sections of themes and subthemes. Theme development was data-driven and closely reflected participant responses [19, 20]. Remaining potential themes were then critiqued by the four authors within two separate phases to further the ideals of the data. Due to the brief period of the study abroad program, the participants could not give feedback on themes and conclusions. The authors collaborated in-depth to analyze each specific theme and subtheme to incorporate varying levels of value. Any remaining issues were resolved through consensus and reviewed until the data were decided upon unanimously.

## Research participants

All participants identified themselves as cisgender women (n = 28, 100%). The average age for interviewees was 37.4 years (SD = ±11.5, range = 25–60 years). Twenty-six participants (92.8%) resided in Florence throughout the study. A certain percentage of participants stated they were in a relationship with one partner (n = 17, 60.7%). Most participants self-identified as heterosexual (n = 26, 92.8%). Researchers found participants had higher education completed (n = 12, 42.9%), or completed graduate education (n = 12, 42.9%) remaining participants completed high school or less (n = 4, 14.2%). The average age of menarche presented as 12.4 years (SD = ±1.7 years, range 25–60 years). Mean age of menopause among the four participants noted was 50.5±4.3 years. Further demographic information related to participants can be seen in Table 2.

## Inclusivity in global research

In adherence to rigorous reporting standards, the present study utilized the Consolidated Criteria for Reporting Qualitative Research (COREQ) framework to guide the methodological design and reporting practices.

# Results

Overall, three themes emerged from the data on menstruation education, menstrual policies, and menstruation impacts from COVID-19. Themes and subthemes are presented below with representative quotes followed by participant age (e.g., (29)). Additional supporting quotes are presented in Table 3.

### "I say 'Mamma what happened to me?'": Menstruation education

The initiation and depth of menstruation education varied among participants and affected their menarche experiences differently. Most were taught about menstruation before menarche, to a point where they felt they "didn't go into it blind" (27). Their anticipation of menarche ranged from "it was not something that was unexpected" (58) to "nonetheless, I was surprised" (60). Additionally, a few participants were informed menarche meant they were growing up and entering puberty. One participant's mother defined menarche as "*Diventare Signorina*, which means. . . you're not a child anymore. You are a *signorina*, a little lady" (60). Not all participants were educated at the same time in their lives, which prepared them for menarche to different degrees.

Other participants reached menarche with no prior knowledge. Another participant, when illustrating her menarche experience, said "they don't have to speak about this. I say, 'Mamma what happened to me?'" (56), and another expressed "they only told me that it was normal, that you were becoming a woman" (37). Participants also explained how more reserved families discussed the topic, with one participant claiming, "no one tell[s] me because my family is

**Table 2. Participant demographics.**

| Participant Characteristic | Distribution of Study |
|---|---|
| **Age (years)** | |
| Age of participants | 37.4 ± 11.5 (range = 25–60) |
| Age of participants in menopause | 50.5 ± 4.3 |
| **Gender Identity** | |
| Woman (cisgender) | 28 (100) |
| Man (transgender) | 0 (0) |
| Gender queer or gender non-conforming | 0 (0) |
| Non-binary | 0 (0) |
| A different identity | 0 (0) |
| Preferred not to answer | 0 (0) |
| **Relationship Status** | |
| Single | 11 (39.3) |
| One partner | 17 (60.7) |
| **Sexual Orientation** | |
| Straight/heterosexual | 26 (92.8) |
| Bisexual | 1 (3.6) |
| Preferred not to answer | 1 (3.6) |
| **Health history** | |
| History of pregnancy | 11 (39.3) |
| Age at menarche (years) | 12.4 ±1.7 |
| **Highest level of education obtained** | |
| High school | 4 (14.2) |
| Undergraduate education | 12 (42.9) |
| Graduate education | 12 (42.9) |
| **Race and ethnicity** | |
| Identified as a person of color | 1 (3.6) |
| Did not identify as a person of color | 27 (96.4) |

Note: Data presented as Avg ± SD or n(%). Numbers that do not add to 100% reflect missing data. The age range is presented alongside the mean and standard deviation.

one of the traditional, extremely old traditions" (56). Some participants also expressed feeling unprepared due to a lack of knowledge. A participant recalled "it was not easy for me" (55), when discussing her experience with irregular periods. The extent to which participants were educated varied widely in timing and content.

Most participants reported having received at least some of their menstruation education from family. One participant explained she was taught "through our parents and our families" (28). Another participant learned from her father, who told her "Don't be scared because it's normal. It happens in every woman" (26). Others noted peers close to them provided menstruation education. One participant said, "what I really know, it comes from peers, the conversations with friends" (30). Each source brought the participants a different depth of knowledge and conversational experience.

Many participants attributed most of their education to what they learned from experiences. One participant explained she would "try to take information from each person you meet, even without asking, just observing and just trying to understand what they're talking about" (37). Another disclosed she was first exposed to menstruation through observing her mother because

**Table 3. Theme, subtheme, and exemplar quotes.**

| Theme | Subtheme | Exemplar Quotes |
|---|---|---|
| **Prevalence of Menstruation Education** | Introduction to Menstruation | "Explained me what it was, I didn't really understand it because I was too young and I didn't know anything about it" (30) |
| | Sources of Education | "In a super soft way, much more from a brother point of view" (28)<br>"So good that I always encourage my friends" (55).<br>"Felt that she was understanding, and she gave me all the information I needed" (58) |
| | Forbidden | "Biology during the high school, maybe they tell you other reproductive things happen, but not really specific on menstruation and menopause" (39)<br>"We need to teach to male[s] . . . because we are having our period doesn't mean that females are weak" (28). |
| **Menstrual Leave Policy** | Yes, to Policy | "because of our national labor system, it should be national. Otherwise, it wouldn't work. The law must be the national law that gives us some procedures, and regulations otherwise it wouldn't work, and it would not be respected" (55).<br>"I guess it makes sense if it's coming from above so maybe national level otherwise everyone's doing their own thing and there's no consistency" (48). |
| | Put the "Me" in Menstrual Leave | "Yeah, a few times, years ago, it happened. But I had a boss, she suffered from endometriosis. So, she understood" (28). |
| **Impacts of COVID-19** | Positive and Negative Impacts on Menstruation | "Everything was delayed. It was very difficult to have an appointment and the appointment was for months later. So big issue" (123,58).<br>"Even now I haven't doctor since two or three years" (48). |
| | No Impact on Menstruation Due to COVID-19 | "I had COVID-19, and nothing happened to me honestly. I didn't have any impact of the period" (37). |

When we go to the bathroom, the door was always open. And so, when I was really young, like six or eight years old, and asked to help, "what happened? Why do you have some blood?" And she explains [to] me (45).

Some participants stressed they learned superstitions such as "there's this kind of mentality when you have menstruation you have to stay in your house, not to speak about that. You cannot cook, you cannot do nothing because trouble can come" (29). No education standard was mentioned in the interviews; therefore, participants noted receiving varied menstruation education approaches and were ultimately left with gaps in their knowledge surrounding the topic.

Participants pointed out how people who do not menstruate are not educated on the topic. One participant noted, "in Italy, the family or the school and the community, don't really talk with the non-menstruators about this topic" (26). Another person added non-menstruators "are scared about it . . . they have to know that it's something that is normal" (28). Most of the participants interviewed expressed some non-menstruators in their lives lacked menstruation knowledge:

I still feel like the male population is really lacking information concerning pads. I've sent my boyfriend. . . to buy pads. He came home with like 50, I don't know, like four huge boxes. I'm like, what is it? One year? He bought me like, which one do you need? He doesn't know, and he's lived with women before. So, I think he's really lacking resources (25).

Most participants agreed menstruation education should be taught to everyone, including non-menstruators, with one participant reinforcing this thought by saying "the girls should know and not only the girls but guys as well" (48). Additionally, one participant said, "at the school, nobody told them. I think that there is a little bit of taboo" (48). Participants believed everyone should be educated on menstruation to minimize negative stereotypes and topic taboos.

Sometimes, the education surrounding menstruation did not meet participants' expectations. Within school, most participants felt the topic of menstruation was "forbidden to mention," and "kept secret [because] it was something that only the women had to manage" (26). Multiple participants explained they studied the reproductive system, but it was "too much technical. So, you understand the bodies, the anatomy, but you don't understand what is the anatomy inside of you" (29). In a few participants' families, discussions of menstruation were vague. This absence of communication persists today as one participant explained her friend's opinion on passing down this information to her daughter as "no, not me. Why should I tell her?" (26). Participants differed on how satisfied they were with their education and how much education they felt should be provided from various sources.

### "It's like taking care of your mental health": Menstrual leave policy

The interviews revealed most participants supported a menstrual leave policy, with one participant stating, "of course, it should. There should be something that, like when you have the flu, you can stay home. Why when you have the period you don't have to?" (27). Another participant added, "in Spain, they are giving three days off to women for periods. . . I hope one day they will do the same in Italy" (37). Many supported menstrual leave because they considered menstruation a part of general wellbeing. One participant stated:

> It's like taking care of mental health. . .It's part of your health, so I think it's very important to also be careful with this in general. So, if you don't feel safe, you don't feel comfortable staying at work or school during that period, I think it's totally fine and good to take a rest or to go home, to try to recover in a way (28).

Additionally, one participant stated, "I think policymakers should come up with this. . . I think that it's something that institutions should make" (27). Another participant echoed this by declaring, "definitely the government" (30). Even though most participants supported the idea, some voiced concerns. A few participants felt unsure about the number of days a menstrual leave policy would allow, saying "I think it's tricky. . . Italians are very bad at this thing because if you give them a finger, they will definitely take the whole hand" (34). A different participant mentioned, "a day fine. . . three days is, maybe it's a bit too much" (48). Despite these concerns, most participants supported implementing a menstrual leave policy as a governmental responsibility.

Most interviewees shared experiences leaving work or school because of menstruation. One menstruator said, "I remember having skipped a few days of school when I was maybe 15 or 16 or 17 in high school. Cause I was feeling really poorly" (25). Another participant shared a similar experience: "it happened [in] school when I was. . . 13, 14 years old because too much pain, I felt a little sick" (35). Not only did participants miss school, they also missed work: "I have some days. . . working that my second day that is my worst. I prefer to stay at home" (26). Additional participants felt comfortable missing work for menstruation-related reasons if needed. For example, a participant said:

> I personally have never had to take time off of work just because I'm capable. But I am confident that if I needed a few hours, I could go to my human resource department and say hey I'm really feeling crappy like I just need to take a few hours off (27).

There was a subset of participants who expressed they felt comfortable missing school, but not work, for menstruation: "School maybe. Work's not. . . you are an adult. I say no" (48).

Most participants from this study have taken time off, whether from school or work, because of menstruation.

### "There were a little bit differences, but now, it's like before": Impact of COVID-19 on menstruation experiences

Many participants discussed the positive or negative impacts COVID-19 had on their menstruation experiences. Some reported enjoying working from home during their menstruation: "I mean, I was home all the time. So, on this side, it actually helped because I didn't feel the struggle. . .to be painful when you're outside or when you have to go to work" (28). Furthermore, participants disclosed being more comfortable menstruating during the pandemic because they were at home, "I think, if there's change it's for the good because comfy" (30). Contrary to the positive impacts, some experienced irregular periods. When asked if the length of their menstruation got shorter during the pandemic, a participant said, "sometimes people didn't have the period for a month" (28). Participants were also asked about their menstrual health after receiving the COVID-19 vaccine. Some participants discussed experiencing no menstrual side effects from the COVID-19 vaccine, noting "I got all the doses but no changes" (25) and "no, nothing; exactly the same" (45). Other participants mentioned negative impacts, with one participant reporting "many of my friends said that they would skip the cycle or last longer" (28). Another participant expressed how ". . .after a certain dose of vaccine, like I got more pain" (26). COVID-19 impacted menstruators in myriad ways.

The COVID-19 pandemic created challenges with accessing healthcare providers and menstruation products. A participant disclosed it is, "not easy to get products at all" (28). Additionally, participants discussed a pandemic policy that restricted the number of products one could purchase and when they can be purchased by defining menstruation products as non-essential goods. One participant explained:

> Because of COVID, we had the curfew at 10[pm] and you couldn't buy first necessities, pads were not included to have later than 7 pm. But the guy at the grocery store said, 'You have to prove it'. He said, 'You have to prove it'. Because you cannot buy not necessary goods after 7 pm. . .and pads were not necessities (26).

Furthermore, participants mentioned during lockdown that they needed approval to purchase menstrual products. A menstruator elaborated they needed "a certificate to go out, and just we had to write down that we needed to go to the supermarket or the pharmacy" (28). Another obstacle menstruators faced was increased difficulty in receiving healthcare "because you are not able to go to the doctor" (58). Healthcare throughout the pandemic was a challenge for all. Another participant stated: "No one could go and check-up or do visits. I know people, parents of my friend, who died because they didn't get the normal check-up, and no one knew that they developed cancer" (26). Some participants shared they experienced no difficulties: "I would say that I have had no impact at all" (28). There are different aspects to the menstruation experience during the COVID-19 pandemic; as such, menstruators were impacted in various ways, including ease of symptom management and availability of menstrual products.

### Discussion

Researchers completed 28 in-person interviews of menstruators residing near Florence, Italy to understand menstruation education, policies, and experiences before and during the COVID-19 pandemic. Emergent themes described the different methods and extent to which menstruators were taught about menstruation, and how it affected their views of future

education and policies. Results further explored participants' positive and negative menstruation experiences during the COVID-19 pandemic. Understanding how menstruators learned about menstruation before and during the COVID-19 pandemic is imperative for identifying effective practices for mitigating personal, social, and cultural burdens among the Italian population. There were no apparent differences between age groups education, experiences, and opinions on menstruation.

Most participants received minimal menstruation education before reaching menarche, and as a result, felt unprepared for the experience. This preliminary education rarely exceeded a simple mention of one day they would bleed and that it is normal. Participants felt unprepared for reaching this milestone and recalled experiencing emotions such as fear and confusion, further enforced through familial values. The preservation of these stigmatizing values caused participants to avoid early menstruation discussions, thus limiting timely education, which is necessary for physical, mental, and social well-being [11]. Many participants noted becoming afraid of their bodies. They were too uncomfortable to discuss their health with someone, especially if they were not educated on what was happening to them during menstruation, which aligns with past work [21]. Additionally, participants shared the lack of menstruation knowledge that non-menstruators in their lives possessed. Menstruation education is not standardized for those who do not menstruate in Italy, with some receiving rudimentary education while others receive none. For some, this results in misconceptions and the creation of taboo feelings surrounding menstruation [21]. Participants in our study agreed everyone, menstruators and non-menstruators, needs to receive menstruation education to reduce stigma and develop a supportive menstruation community culture. To improve health and overall well-being, changes to sexual health education should be made to include information about menstruation before menarche, beyond the simple fact that one day menstruators will bleed [22], and involve non-menstruators to educate them on this subject properly.

Numerous participants were unaware if Italy had a menstrual leave policy, but felt there should be one in place, aligning with previous research and noting policy support [6]. One common occurrence was taking off work or school due to negative menstrual symptoms, but these absences were categorized as sick leave of absence due to no existing menstrual policy. This leaves them feeling frustrated because they have no security to take off days due to menstruation. A menstrual leave policy could eliminate these insecurities and allow menstruators to feel heard, valued, and not set back after taking care of their menstrual health. While most supported this policy, some voiced concerns about it becoming abused. Subsequently, menstruators may feel uncomfortable enough to utilize this policy due to perceptions of too much time away from work, or being perceived as unable to complete high-quality work while menstruating. Participants believe a related policy would be respected and taken seriously if led by a governmental official or agency. Nonetheless, menstruators should be taught about menstrual leave policies and how to discuss accommodations with supervisors.

Many participants noted COVID-19 didn't affect their menstruation, while some said it had a positive impact and others reported negative impacts due to the pandemic. Among the participants who experienced adverse effects, a commonly disclosed symptom was cycle irregularity. Takmaz et al. [13] reported inconsistent cycles are correlated with high stress, and COVID-19 was reported as a stressor for many menstruators. Furthermore, while most participants did not notice any side effects related to their menstruation after receiving their COVID-19 vaccination, those who did noted an increased length of pain during their period. This aligns with previous research concluding there was a correlation between the vaccine and irregular menstrual cycle lengths as well as quantity of blood [14].

A quarter of European menstruators endure negative impacts from a heavy flow, resulting in decreased work capacity [23]. Prior to COVID-19, menstruators who experienced painful

or heavy flows missed work due to their symptoms. However, many participants noted that during the COVID-19 pandemic, they could work from home, resulting in a more comfortable menstruation experience and greater flexibility to manage symptoms. Furthermore, many menstruators preferred to work from home during their period because it allowed them to dress casually, improving comfortability. Additionally, menstruators had trouble purchasing menstrual products during the COVID-19 pandemic due to quarantine restrictions, as items such as pads and tampons were deemed non-essential goods. Understanding the impacts, the COVID-19 pandemic had on menstruators can improve menstruation policies and education.

## Implications

The lack of menstruation education can leave menstruators and even non-menstruators at a disadvantage by the time menarche occurs. Educators, policymakers, and schools in Italy should work to standardize the menstruation curriculum better to suit this population's information and timing needs. Misinformation and misconceptions regarding reproductive health issues, including menstruation, prevent menstruators from seeking help and guidance when it may be needed. Menstruators deserve to feel supported in their schools, homes, workplaces, and communities. Quarantines during the COVID-19 pandemic relieved much stress for menstruators, as it reduced the inclination to prioritize certain societal expectations, like formal wardrobe, and provided more comfortable remote work environments. A menstrual leave policy allowing working menstruators monthly paid time away from work could facilitate greater prioritization of menstrual health. Policymakers should consider the attitudes and opinions towards menstrual leave to use as support for crafting a policy that better supports menstrual health in Italy.

## Strengths, limitations, and further research

A team of 28 all-female, trained interviewers with experience in research methodology conducted the interviews. Researchers spent roughly two months in the country, allowing them time to fully immerse themselves in the culture and familiarize themselves with the community in Florence, Italy. Amidst an all-female team, the participants may have been more comfortable speaking about topics relating to menstruation. Although the interview experiences varied among the team members, the nature of the research team incorporated an open environment, encouraging the sharing of insight between team members. While there are many strengths in this research, some limitations still exist. In this study, participants were recruited if they could speak conversational English. This potential bias limited the scope of participants to choose from and could have affected the range of experiences and opinions collected. As all participants shared the ability to speak English, this could indicate common backgrounds, values, experiences, or education shared that narrow down the variety in our data sample. If comfortable, the participants then choose whether to proceed after a brief introduction of what the study and interview entailed. Apart from speaking conversational English, participants were recruited within or around Florence, Italy, potentially making results not generalizable demographically or geographically to other populations. While this study had a wide range of ages and educational levels, most of the participants were heterosexual, and all were cis-gendered females, resulting in minimal sexual diversity in our data.

Future studies should include individuals who self-identify as unemployed, uneducated, and who are not limited to conversational English to understand a broader range of menstruation experiences. Conducting methodology in Italian and English will ameliorate confusion around language barriers regarding policy, COVID-19, and education. Furthermore, future interviews could include specific questions to elaborate further on the connections between

these topics. Finally, studies should incorporate interviews from other geographic areas to encompass a fuller scope of European menstruators.

## Conclusion

Results found most menstruators learned about menstruation through their family before experiencing menarche. Regardless, menstruators did not consider themselves prepared for their first cycle. Analysis revealed a majority of participants had experienced missing school or work due to menstruation, leading participants to believe it would be beneficial to implement a menstrual leave policy. In addition, the results conveyed the positive and negative impacts COVID-19 had on menstruators in Florence, Italy. Findings demonstrated challenges menstruators faced, including decreased access to menstrual supplies and symptom management.

## Supporting information

**S1 File. Interview guide.**
(DOCX)

**S1 Checklist. Inclusivity in global research.**
(DOCX)

## Acknowledgments

We thank the students who participated in the Summer 2022 Purdue University Investigating Women's Reproductive and Sexual Health Issues in Florence, Italy, study abroad program for supporting data collection, transcription, and overall collaboration on the project. We would also like to thank our Florence University of the Arts colleagues for their partnership and project support.

## Author Contributions

**Conceptualization:** Meagan R. DeMark, Grace A. Khamis, Megan S. Rach, Jaslin A. Martinez, Andrea L. DeMaria.

**Data curation:** Meagan R. DeMark, Grace A. Khamis, Megan S. Rach, Jaslin A. Martinez, Andrea L. DeMaria.

**Formal analysis:** Meagan R. DeMark, Grace A. Khamis, Megan S. Rach, Jaslin A. Martinez, Andrea L. DeMaria.

**Funding acquisition:** Andrea L. DeMaria.

**Investigation:** Meagan R. DeMark, Grace A. Khamis, Megan S. Rach, Jaslin A. Martinez, Andrea L. DeMaria.

**Methodology:** Meagan R. DeMark, Grace A. Khamis, Megan S. Rach, Jaslin A. Martinez, Andrea L. DeMaria.

**Project administration:** Meagan R. DeMark, Grace A. Khamis, Andrea L. DeMaria.

**Resources:** Andrea L. DeMaria.

**Software:** Andrea L. DeMaria.

**Supervision:** Andrea L. DeMaria.

**Validation:** Andrea L. DeMaria.

**Visualization:** Andrea L. DeMaria.

**Writing – original draft:** Meagan R. DeMark, Grace A. Khamis, Megan S. Rach, Jaslin A. Martinez.

**Writing – review & editing:** Meagan R. DeMark, Grace A. Khamis, Megan S. Rach, Jaslin A. Martinez, Andrea L. DeMaria.

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
