## [Decision Letter · Decision Letter 0]

12 Feb 2024

PGPH-D-23-02370

“You have to stay in your house…because trouble can come”: The impact of education, policy, and COVID-19 on menstruation experiences in Florence, Italy

Dear Dr. DeMaria,

Thank you for submitting your manuscript to PLOS Global Public Health. After careful consideration, we feel that it has merit but does not fully meet PLOS Global Public Health’s publication criteria as it currently stands. Therefore, we invite you to submit a revised version of the manuscript that addresses the points raised during the review process.

Please revise your manuscript in response to the reviewer's comments.

We look forward to receiving your revised manuscript.

Kind regards,

Anil Gumber, Ph.D.

Academic Editor

Journal Requirements:

2. Please ensure that Funding Information and Financial Disclosure Statement are matched.

3. In the Funding Information you indicated that no funding was received. Please revise the Funding Information field to reflect funding received.

4. We have noticed that you have uploaded Supporting Information files, but you have not included a list of legends. Please add a full list of legends for your Supporting Information files after the references list.

5. In the online submission form, you indicated that "The dataset used and analysed for the current study are available from the corresponding author upon reasonable request". All PLOS journals now require all data underlying the findings described in their manuscript to be freely available to other researchers, either 1. In a public repository, 2. Within the manuscript itself, or 3. Uploaded as supplementary information.

Additional Editor Comments (if provided):

Reviewers' comments:

Reviewer's Responses to Questions

**Comments to the Author**

1. Does this manuscript meet PLOS Global Public Health’s publication criteria? Is the manuscript technically sound, and do the data support the conclusions? The manuscript must describe methodologically and ethically rigorous research with conclusions that are appropriately drawn based on the data presented.

Reviewer #1: Yes

Reviewer #2: Partly

2. Has the statistical analysis been performed appropriately and rigorously?

Reviewer #1: N/A

Reviewer #2: N/A

3. Have the authors made all data underlying the findings in their manuscript fully available (please refer to the Data Availability Statement at the start of the manuscript PDF file)?

Reviewer #1: No

Reviewer #2: Yes

4. Is the manuscript presented in an intelligible fashion and written in standard English?

Reviewer #1: Yes

Reviewer #2: Yes

5. Review Comments to the Author

Reviewer #1: This research is relevant to understand how unprecedented situations like COVID-19 pandemic would influence menstruators in terms of their work and accessibility to menstrual products. The researchers have also tried to seek details on menstruators' awareness at menarche.

The methodology is quite robust and reliability of the data is ensured by the PI. A sample of 28 participants is quite adequate to give insights into diverse experience, given the geographical boundaries within which the study is undertaken. Content analysis is also undertaken meticulously.

However, it is suggested that the authors address the following concerns so that whatever little gaps that appear in the manuscript can be filled in:

1. The researchers have undertaken snowball sampling to interview 28 participants for this survey. It would be good if the authors could explain in a line or two about the base sample selection (from whom snowballing was done).

2. The authors mention that all the participants whose interviews were audio-recorded signed an informed consent form. Was informed consent not sought from all the participants, who were interviewed? If not, give an explanation why? If yes, please mention explicitly how the consent was sought, with or without audio-recording. Informed consent is important irrespective of audio-recording or not.

3. The phrase "education, policy and COVID-19 concerning menstruation" has been used several times in the manuscript. Do the authors mean, "awareness at menarche, menstruation leave policy and the restrictions associated with COVID-19"? It would be good to clarify this point, since this is the very objective of the study and that requires to be clearly and unambiguously stated.

Other comments are made at relevant places in the manuscript (uploaded herewith).

Reviewer #2: As a reviewer, I congratulate the authors for presenting their research concerning a topic that requires holistic and multilevel intervention from policymakers and public health institutions worldwide. When reviewing the paper, I came across some questions and doubts that arose during the process:

In the abstract, there needs to be more clarity regarding what the authors mean by community standards, either standards of education or standards of care and awareness.

Also in the abstract, the statement: "Current menstruation education minimally supports menstruators and other European countries have started employing menstrual policies" is confusing to follow for the readers since there is no information regarding the context that the authors try to convey.

In the purpose of the study presented in the abstract, there needs to be more clarity regarding the impact of COVID-19 that the authors try to investigate, whether it impacts overall well-being, menstruation patterns and symptoms, or life quality.

It is recommended that the authors explicitly state that their study is limited to a specific population within a city (Florence) rather than generalizing about "menstruators living in Italy."

Given that the study was conducted in Italy in collaboration with the Florence University of Arts, it is recommended that the authors provide a statement regarding the ethical approval (or no need for it) from the Florence University of Arts.

Did the authors consider the possible bias selection by including only English-speaking participants in a non-native English-speaking population? If so, it is recommended that the authors state how this potential bias affects their results and conclusions.

The authors did not mention a delimited or specified localization for the recruitment of the participants. Was there a specified consensus from the researchers regarding this topic? It is recommended that the authors comment on the selection of the participants and the recruitment site since this could affect the results.

It is recommended that the authors specify the training and requirements of the researchers in charge of data collection since the topics for the interview require specific baseline knowledge for proper guidance. Also, it is recommended that the authors specify what they mean when they mention "graduate-level research methodologies."

The authors mention "through being engrossed in Florentine culture." However, in the results, the authors comment about the briefness of the period the students were in the study abroad program; as well, neither in the methods section nor the results there is no specification of whether one of the researchers was originally from the selected population to state the "engrossment" within the culture.

Is the protocol for the interview publicly available? Or has the questionnaire been previously validated within the bigger project mentioned by the authors?

Does the software used for audio recording during the interviews require a license? It is recommended that the authors make a statement regarding this concern to avoid ethical issues.

The authors state that the depth of knowledge and readiness regarding menstruation among the participants varied. It is recommended that the authors clarify what the outcome definition or measurement is regarding these two concepts.

Given the broad age range among the participants, did the authors modify these questions according to age? Did the authors identify and search for patterns in the participants' responses in different age groups?

It is recommended that the authors specify what their measured impact is. Since it is unclear whether this impact affects just the menstruation patterns or if it also includes effects on the familiar environments of the participants. Finally, the authors present "there were varying levels of impact"; however, there is no previous explanation as to what these levels might be or as to how the impact was measured since it is presented in levels.

The authors previously stated the need for education regarding menstruation to be imparted to “boys, men and menstruators” however, in the implications there is a statement that can be interpreted as contradictory since it is suggested that the education should be offered only for “menstruators” since the lack of the mentioned education leaves them at disadvantage.

It is recommended that the authors clearly stated their strengths as well as their limitations, including as to how the authors interpret that the predominant sexual orientation (mentioned as “most of the participants were heterosexual”) presented a possible limitation in interpreting the results.

It is recommended that the authors unify the language to use in their paper, since the participants are described as “menstruators” while when describing the need for education regarding menstruation topics, the authors mention “boys and men”.

6. PLOS authors have the option to publish the peer review history of their article (what does this mean?). If published, this will include your full peer review and any attached files.

**Do you want your identity to be public for this peer review?** For information about this choice, including consent withdrawal, please see our Privacy Policy.

Reviewer #1: **Yes: **Smruti Bulsari

Reviewer #2: No

---

## [Decision Letter · Decision Letter 1]

17 May 2024

PGPH-D-23-02370R1

“You have to stay in your house…because trouble can come”: The impact of education, policy, and COVID-19 on menstruation experiences in Florence, Italy

Dear Dr. DeMaria,

Thank you for submitting your manuscript to PLOS Global Public Health. After careful consideration, we feel that it has merit but does not fully meet PLOS Global Public Health’s publication criteria as it currently stands. Therefore, we invite you to submit a revised version of the manuscript that addresses the points raised during the review process.

We look forward to receiving your revised manuscript.

Kind regards,

Miquel Vall-llosera Camps

Staff Editor

Journal Requirements:

Additional Editor Comments:

Reviewer #2 has raised concerns about the use of inclusive language in this study and we share the concern with the reviewer. This is a qualitative study about menstruation experiences in Italy. The words 'woman' or 'women' are only used in the quotes from the participants, however the words 'boys' and 'men' were used until they were replaced by 'non-menstruators', after the reviewer noticed this inconsistency (although not completely, see line 336). We have also noticed that there is no breakdown of participants other than by sexual orientation. Please provide (1) the gender/sex breakdown of the participants, and (2) justification for the use of 'menstruators' instead of 'women' in this context.

Reviewers' comments:

Reviewer's Responses to Questions

**Comments to the Author**

1. If the authors have adequately addressed your comments raised in a previous round of review and you feel that this manuscript is now acceptable for publication, you may indicate that here to bypass the “Comments to the Author” section, enter your conflict of interest statement in the “Confidential to Editor” section, and submit your "Accept" recommendation.

Reviewer #1: All comments have been addressed

Reviewer #2: (No Response)

2. Does this manuscript meet PLOS Global Public Health’s publication criteria? Is the manuscript technically sound, and do the data support the conclusions? The manuscript must describe methodologically and ethically rigorous research with conclusions that are appropriately drawn based on the data presented.

Reviewer #1: Yes

Reviewer #2: Partly

3. Has the statistical analysis been performed appropriately and rigorously?

Reviewer #1: N/A

Reviewer #2: N/A

4. Have the authors made all data underlying the findings in their manuscript fully available (please refer to the Data Availability Statement at the start of the manuscript PDF file)?

Reviewer #1: No

Reviewer #2: Yes

5. Is the manuscript presented in an intelligible fashion and written in standard English?

Reviewer #1: Yes

Reviewer #2: Yes

6. Review Comments to the Author

Reviewer #1: (No Response)

Reviewer #2: As a reviewer, I would like to thank the authors to address most of the suggestions and comments.

7. PLOS authors have the option to publish the peer review history of their article (what does this mean?). If published, this will include your full peer review and any attached files.

**Do you want your identity to be public for this peer review?** For information about this choice, including consent withdrawal, please see our Privacy Policy.

Reviewer #1: **Yes: **Smruti Bulsari

Reviewer #2: No

---

## [Editor Report · Decision Letter 2]

14 Jun 2024

“You have to stay in your house…because trouble can come”: The impact of education, policy, and COVID-19 on menstruation experiences in Florence, Italy

PGPH-D-23-02370R2

Dear Dr. DeMaria,

We are pleased to inform you that your manuscript '“You have to stay in your house…because trouble can come”: The impact of education, policy, and COVID-19 on menstruation experiences in Florence, Italy' has been provisionally accepted for publication in PLOS Global Public Health.

Before your manuscript can be formally accepted you will need to complete some formatting changes, which you will receive in a follow up email. A member of our team will be in touch with a set of requests. Please ensure that you remove the footnote and integrate the text into the paper itself, as footnotes are not allowed per our formatting requirements.

Best regards,

Julia Robinson

Staff Editor